# Effects of Shared Decision-Making, Health Literacy, and Self-Care Knowledge on Self-Care Behavior Among Hemodialysis Patients in Korea: A Cross-Sectional Survey

**DOI:** 10.3390/healthcare13020175

**Published:** 2025-01-17

**Authors:** Hyohjung Lee, Mi-Kyoung Cho

**Affiliations:** Department of Nursing Science, Research Institute of Nursing Science, School of Medicine, Chungbuk National University, Cheongju 28644, Republic of Korea

**Keywords:** renal dialysis, self-care, health literacy, health knowledge, decision-making, shared

## Abstract

Background: Patients undergoing hemodialysis for chronic kidney failure experience various complications and physical and emotional difficulties, leading to decreased quality of life. Self-care behaviors are essential for preventing complications and reducing mortality rates. Effective self-care behaviors significantly depend on shared decision-making, health literacy, and self-care knowledge, each critical in patient self-care performance and disease management. This study aimed to determine the importance and relevance of shared decision-making, health literacy, and self-care knowledge. In particular, it enhances self-care behaviors among hemodialysis patients. Methods: Participants were 108 adult hemodialysis patients from the hemodialysis centers of three medical institutions in Cheongju City, Korea. Moreover, the study utilized a descriptive survey research design. Data were analyzed using descriptive statistics, independent *t*-tests, one-way ANOVA, Pearson’s correlation coefficients, and multiple linear regression analyses. Results: The average score for self-care behaviors was 125.28 out of 175 points, with high scores for medication and vascular management. Furthermore, there were low scores for social activities and dietary management. Factors influencing self-care behaviors in hemodialysis patients were identified as sex, age, economic status, health literacy, and self-care knowledge. These factors explained 45.2% of the variance in self-care behaviors. Conclusions: To promote self-care behaviors in hemodialysis patients, it is essential to improve health literacy and self-care knowledge, strengthen tailored educational programs, and promote the explanatory role of nurses and shared decision-making. Additionally, comprehensive interventions, including economic support, are necessary.

## 1. Introduction

Hemodialysis is a renal replacement therapy (RRT) for end-stage renal disease (ESRD) patients. When ESRD occurs, dialysis or kidney transplantation is essential to restore kidney function [1]. Globally, the number of patients with ESRD requiring RRT continues to rise [2]. According to Korean statistics published in 2021, the incidence and prevalence of ESRD have steadily increased over the past 30 years, with hemodialysis accounting for approximately 83.6% of all RRT modalities [3].

Patients undergoing hemodialysis are vulnerable to serious complications such as uremia, electrolyte imbalances, anemia, and heart failure [4]. They continuously experience physical and emotional symptoms, such as fatigue, pruritus, and anxiety [5]. Additionally, hemodialysis increases the economic burden and prolongs treatment time, restricting social activities and employment, ultimately reducing these patients’ quality of life [6,7]. Therefore, engaging in self-care behaviors is crucial to reduce complications and improve quality of life. These behaviors—including dietary control, exercise, and medication management—are important for maximizing treatment efficacy and minimizing complications [8,9,10,11].

One important factor that promotes self-care behaviors is shared decision-making between patients and healthcare providers [12]. Shared decision-making is a collaborative process in which patients and medical staff exchange information and make important health-related decisions. This interaction helps patients understand their treatment options and develop positive attitudes toward treatment [13,14]. The major barriers to effective shared decision-making include patients’ lack of knowledge [15] and insufficient health literacy [16]. Hemodialysis patients often have multiple chronic conditions that improve their health literacy [17], which is essential for promoting self-care behaviors and facilitating shared decision-making [18]. Health literacy is the ability to access, understand, and apply information necessary for health management, disease prevention, and health promotion, and plays a crucial role in maintaining or improving patients’ health and quality of life [19]. Studies have shown a positive correlation between health literacy and self-care behaviors in hemodialysis patients [20]. Low health literacy levels can lead to a lack of self-care knowledge and decreased medication adherence. Failure to adhere to treatment plans may increase healthcare utilization and mortality [21].

Self-care knowledge refers to the information individuals need daily to maintain and promote well-being throughout their lifespan [22]. Sufficient self-care knowledge is essential for patients to perform key self-care behaviors—such as medication adherence [23], dietary control [24], and exercise [25]—thus improving treatment outcomes [26].

This study aimed to identify the effects of shared decision-making, health literacy, and self-care knowledge on the self-care behaviors of hemodialysis patients. Through this study, we intend to contribute to the development of evidence-based nursing interventions aimed at improving self-care behaviors in hemodialysis patients.

## 2. Materials and Methods

### 2.1. Study Design

This study is a descriptive survey designed to identify the relationships of health literacy, self-care knowledge, and shared decision-making on the self-care behaviors of patients undergoing hemodialysis.

### 2.2. Study Population and Sampling

This study was conducted on patients undergoing hemodialysis after being diagnosed with chronic renal failure in the hemodialysis centers of three medical institutions in C city, Korea, In Korea, healthcare facilities are categorized into tertiary hospitals, general hospitals, and clinics. Selecting participants from each category was deemed an effective approach to ensuring a representative sample that reflects the national healthcare system. Accordingly, these institutions were specifically chosen to enhance the study’s representativeness across different levels of healthcare care. The specific inclusion criteria were as follows: (1) adults aged 19 years or older; (2) patients diagnosed with end-stage renal disease who had been receiving hemodialysis therapy for at least three months; (3) patients who visited the outpatient clinic at least twice a week to undergo hemodialysis; (4) individuals capable of reading Korean and responding to the questionnaire; and (5) participants who understood the purpose and methods of this study and agreed to participate. Participants were excluded if they wished to withdraw; had cognitive, physical, or mental complications requiring specialized treatment; experienced difficulties in performing self-care; or had previously participated in a self-care promotion program. Upon receiving approval from the administrative and medical staff, the chief nursing officer informed potential participants about the study. Interested individuals were further educated on the study’s objectives, methods, benefits, and potential risks by the researchers. Written informed consent was obtained from all participants who agreed to partake in the study. A total of 108 participants were enrolled without any dropouts. As a gesture of appreciation, participants received a small token of thanks, and their responses were collected and securely stored by the researchers.

The required number of study participants was calculated using the sample size calculation program G*Power 3.1.9.7 (Heinrich-Heine-Universität, Düsseldorf, Germany) [27]. The sample size was determined based on a medium effect size (0.30 [28], a significance level of 0.05, a power of 0.90, and 17 predictor variables (14 items related to participant characteristics and items on shared decision-making, health literacy, and self-care knowledge). Considering a dropout rate of 10%, 108 participants were selected, and all 108 questionnaires were analyzed without any dropouts.

### 2.3. Measurements

#### 2.3.1. Participants Characteristics

Seven items—sex, age, education level, religion, presence of an assistant, economic activity status, and economic condition—were used to measure the participants’ general characteristics. Disease-related characteristics were assessed using seven items measuring experience with hemodialysis education, the frequency of hemodialysis, the duration of hemodialysis, time required for hemodialysis, causes of end-stage renal disease, the number of chronic disorders other than kidney disease, and the number of medications taken.

#### 2.3.2. Self-Care Behavior

Self-care behaviors in hemodialysis patients were assessed using a self-care behavior tool developed for hemodialysis patients by Cho [11]. This tool consists of 35 items divided into the following categories: diet (6 items), vascular management (6 items), exercise and rest (4 items), medication intake (2 items), blood pressure and weight management (3 items), social activities (3 items), and physical care (11 items). Each item is rated on a Likert 5-point scale (ranging from 1 to 5), where higher total scores indicate better performance in self-care behaviors. At the time of its development, the reliability of the tool was 0.86; in this study, it was 0.88.

#### 2.3.3. Shared Decision-Making

Shared decision-making was assessed using the shared decision-making questionnaire (SDM-Q-9) developed by Kriston et al. [29]. This tool comprises nine items, each rated on a 6-point Likert scale (ranging from 1 to 6), with higher total scores indicating a higher degree at the time of development and 0.89 in this study.

#### 2.3.4. Health-Literacy

Health literacy was assessed using the short-form health literacy scale (HLS-SF12) developed by Duong et al. [30]. This tool consists of 12 items divided into three categories: healthcare (4 items), disease prevention (4 items), and health promotion (4 items). Each item is rated on a 4-point Likert scale (ranging from 1 to 4), with higher total scores indicating higher health literacy. The reliability of the tool was 0.87 at the time of its development and 0.91 in this study.

#### 2.3.5. Self-Care Knowledge

Self-care knowledge of hemodialysis was measured using the tool employed by Yu [31] for hemodialysis patients. This tool consists of 15 items, including the functions and characteristics of normal kidneys (6 items), hemodialysis (1 item), diet (3 items), medication (2 items), complications and subsequent management (2 items), and exercise and daily activities (1 item). Responses were given as ‘yes’ or ‘no’, with correct answers scoring 1 point and incorrect answers scoring 0 points. Higher total scores indicated greater knowledge of self-care. The reliability of the tool was 0.76 during its development, and the KR-20 index was 0.70 in this study.

### 2.4. Data Collection and Ethical Considerations

This study collected data from 108 hemodialysis patients in the hemodialysis centers of three medical institutions in Cheongju City, Korea, from 24 September to 4 October 2024. Before data collection, approval was obtained from the institutional review board of the C University (IRB no. CBNU-2024-A-0025). Participants who provided informed consent after understanding the purpose and content of the study completed a questionnaire. Upon completion, participants submitted the sealed envelope questionnaire directly to the researcher. As a token of appreciation, a full-body moisturizer was provided to participants.

### 2.5. Data Analysis

The collected data were analyzed using SPSS for Windows (version 29.0.2; IBM Corp., Armonk, NY, USA). Descriptive statistics were used to analyze the participants’ general characteristics, disease-related characteristics, shared decision-making, health literacy, self-care knowledge, and self-care behaviors. Differences in hemodialysis self-care behavior scores according to the general and disease-related characteristics of the participants were evaluated using independent *t*-tests and one-way analysis of variance (ANOVA), with post hoc analysis performed using the Scheffé test. Pearson’s correlation coefficients were used to identify the relationships between variables, and multiple linear regression analysis was conducted to evaluate the factors influencing self-care behaviors during hemodialysis.

## 3. Results

### 3.1. Participant’s Characteristics

The average age of the participants was 61.99 ± 13.81 years, with a nearly equal distribution of men (n = 56, 51.9%) and women (n = 52, 48.1%). The most common educational level was 10~12th grade (n = 41, 38.0%), and most reported receiving daily living support from their spouses or families (n = 102, 94.4%). Seventy-six participants (70.4%) reported no employment, and the majority felt that their economic status was moderate (n = 64, 59.2%). All participants received dialysis education, and the majority (n = 106, 98.1%) underwent dialysis three times per week. The average duration of dialysis was 6.53 ± 5.30 years, and the dialysis time was consistently 4 h for all. The primary causes of end-stage renal failure were diabetes mellitus (n = 57, 52.8%) and hypertension (n = 26, 24.1%; Table 1).

### 3.2. Descriptive Statistics of the Variables

The self-care behaviors had an average score of 125.28 ± 16.25 out of 175 points. Subscores were as follows: medication intake scored 9.39 ± 1.13, vascular management scored 25.83 ± 3.70, physical care scored 43.85 ± 5.57, exercise and rest scored 13.93 ± 3.27, blood pressure and weight management scored 9.74 ± 3.36, dietary management scored 18.60 ± 3.90, and social activities scored 7.00 ± 2.85. The shared decision-making had an average score of 45.65 ± 6.19 out of a total of 54 points, and health literacy had an average score of 34.36 ± 7.85 out of 48 points. Self-care knowledge scored an average of 12.86 ± 2.10 out of 15 points (Table 2).

### 3.3. Differences in Hemodialysis Self-Care Behaviors by Participant Characteristics

The normality of the data was confirmed before analysis using the Kolmogorov–Smirnov test. The results showed statistically significant differences in hemodialysis self-care behaviors based on sex, age, and economic status. Women scored higher than men (t = −2.39, *p* = 0.019). Those aged 55–70 years scored higher than those aged < 55 years, and those aged > 70 years scored higher than those aged 55–70 years (F = 6.53, *p* = 0.002). Participants with sufficient economic status scored higher than those with moderate or insufficient economic status (F = 11.92, *p* < 0.001; Table 3).

### 3.4. Correlations Among the Variables

In this study, a significant positive correlation was found between hemodialysis self-care behaviors and shared decision-making (r = 0.28, *p* = 0.003). There was also a positive correlation with health literacy (r = 0.42, *p* < 0.001), and self-care knowledge showed a significant correlation as well (r = 0.29, *p* = 0.003; Figure 1).

### 3.5. Factors Influencing Hemodialysis Self-Care Behaviors

In this study, multiple regression analysis was used to investigate the factors influencing self-care behaviors in patients undergoing hemodialysis (Table 4). The regression results identified sex, age, economic status, health literacy, and self-care knowledge as significant influencing factors. Self-care behavior levels were significantly higher in women compared to men (β = −0.26, *p* = 0.001) and increased with age (β = 0.36, *p* < 0.001). Economic status also showed that those with sufficient resources had significantly higher levels of self-care behaviors compared to those with insufficient resources (β = 0.25, *p* = 0.006). Additionally, higher health literacy (β = 0.41, *p* < 0.001) and greater self-care knowledge (β = 0.17, *p* < 0.001) were associated with significantly higher levels of self-care behaviors. These variables explained 45.2% of the variance in self-care behaviors among hemodialysis patients (F = 13.59, *p* < 0.001).

## 4. Discussion

In this study, the average score for self-care behaviors among hemodialysis patients was 3.68, which is higher than the 3.34~3.52 reported in previous studies [8,10,32]. In the self-care behavior categories, high scores were observed for medication management (4.69), vascular management (4.31), and physical activity (3.99), indicating that patients focused on categories where they could directly experience therapeutic effects [8]. In contrast, low scores on social activities (2.33), dietary management (3.10), and blood pressure and weight management (3.25) indicated deficiencies in self-care behaviors in these categories [8,10]. This finding indicates that hemodialysis patients find it difficult to participate in social activities because of time constraints and physical fatigue caused by dialysis, and they struggle with continuous self-management, such as dietary control and blood pressure management [5]. Therefore, support programs and continuous education that consider the psychological state of patients are necessary to improve self-care behaviors in these categories [33]. In particular, tailored education that strengthens social support networks and raises awareness of the importance of dietary control is required [5].

Factors influencing self-care behavior included sex, age, economic status, health literacy, and self-care knowledge. In this study, female patients exhibited higher levels of self-care behavior than male patients, which aligns with the results reported by Sousa et al. [34]. However, some prior studies have found that males perform self-care behaviors better [7,35]. In contrast, others report no significant differences based on sex [36,37], suggesting the need for further research to clarify the differences in self-care behaviors across specific areas based on sex. Age also had a significant impact on self-care behavior. While some studies have reported higher levels of self-care behavior in younger patients [24,35], others have suggested that younger patients exhibit lower levels [38,39]. In this study, patients aged < 55 years showed lower levels of self-care behaviors, indicating the need to develop educational programs emphasizing the importance of self-care among younger patients. Patients with sufficient economic status exhibited higher self-care behaviors, consistent with previous research [32,37,40], demonstrating that economic difficulties can negatively affect psychological well-being and self-care behaviors. Therefore, policy support is needed to alleviate the economic burden, and strategies are needed to promote psychological stability.

Hayati et al. [41] reported that having someone living with and assisting hemodialysis patients significantly improves self-care behaviors such as dietary habits, fluid restrictions, and medication adherence and can also reduce blood phosphorus and potassium levels. However, in this study, the presence of an assistant did not significantly improve self-care behaviors. This might be due to a potential bias in the analysis of the results, as only six participants reported not having an assistant. Future studies should consider increasing the number of participants and using stratified sampling to address this issue. The frequency of dialysis [35] and dialysis duration [42] have been reported in previous studies as factors that show significant differences in self-care behaviors. In this study, 98.1% of the patients received dialysis three times a week, with each session lasting 4 h. These findings are consistent with reports that 91.7% of dialysis patients in Korea undergo dialysis three times a week for four hours per session [3]. According to a study comparing global dialysis statuses by Lee et al. [43], most countries with available global data receive hemodialysis two to three times per week, each lasting three to four hours. However, Vasquez-Jimenez and Madero [44] reported that the average weekly frequency of hemodialysis in Mexico is 1~2 times, and only 2% of patients receive dialysis 3 times a week. Garcia and Sanchez-Polo [45] noted that most patients in Guatemala receive dialysis once a week and that the frequency varies depending on the type of insurance. Therefore, future studies should adjust these factors to fit each country’s medical environment and patient characteristics.

In this study, health literacy was identified as the most influential factor in the self-care behavior of hemodialysis patients. This aligns with previous studies [20,21], suggesting that higher health literacy improves self-care behaviors. The average health literacy score of the study participants was 2.86, which is lower than the average of 2.98 for the general adult population but higher than the average of 2.70 for hemodialysis patients [46]. These findings indicate the need for improved health literacy among patients undergoing HD. Therefore, it is important to develop customized educational programs that reflect patients’ individual characteristics and needs, and the use of easy-to-understand language and visual materials is essential [47]. Additionally, by enhancing nurses’ educational and counseling capabilities and expanding the use of explanatory nursing systems, patients can receive adequate information about their treatment processes and self-management [48]. Improving health literacy can lead to better self-care behaviors and treatment outcomes through such efforts.

Self-care knowledge was identified as a significant factor influencing self-care behaviors in this study, consistent with previous research [23,49], which suggests that higher self-care knowledge enhances self-care behaviors. Hafezieh et al. [50] reported that self-care knowledge increases patient self-efficacy, promoting self-care behaviors. Xu et al. [23] noted that self-care knowledge improves treatment adherence. The average score for self-care knowledge in this study was 12.86, which is lower than the 13.33 reported by Yu et al. [31], suggesting that participants’ level of self-care knowledge may be relatively low. This indicates the need for additional education and interventions.

Previous studies have reported that experience with hemodialysis education significantly influences self-care behaviors [9,33,51]. However, this study could not analyze differences based on the presence or absence of education since all participants had received training. However, self-care behaviors in this study were higher than in previous studies, suggesting that education may have positively impacted the observed self-care outcomes. Therefore, continuous and systematic education tailored to patients’ changing conditions and individual needs is essential to maintain self-care knowledge and enhance self-care behaviors. Ren et al. [51] reported that education using online videos is effective, and the teach-back method, where patients explain the information themselves to increase understanding, contributes to improving self-care knowledge, self-efficacy, and self-care behaviors [52,53,54]. These educational methods must also be applied in domestic settings, particularly through the development of customized education programs. This approach can enhance self-care knowledge in hemodialysis patients and improve self-care practices, treatment outcomes, and quality of life.

While shared decision-making did not significantly influence self-care behavior in the regression analysis, the correlation analysis showed significant positive correlations with self-care behavior, self-care knowledge, and health literacy. This suggests that shared decision-making may indirectly affect self-care behavior by mediating other factors. Therefore, future research should clarify the indirect effects of shared decision-making and the interactions among variables using mediation analysis or structural equation modeling. This aligns with previous research indicating higher levels of self-care behavior as patients participate more actively in the treatment decision-making process [55,56]. Chung et al. [15] emphasize the central role of nurses in the shared decision-making process. Nurses can assist patients in decision-making by providing the necessary information and support. Additionally, a recent review study [57] reported that nurses’ relational skills play a crucial role in the shared decision-making process by providing emotional support to patients and fostering trust between healthcare providers and patients. These studies highlight the need to develop strategies to promote patient-centered shared decision-making. However, this study was limited to specific regions and participants, which limits the generalizability of the results. Future research should investigate shared decision-making across various healthcare settings and include a broader demographic group to deeply explore the impacts of cultural and racial factors. This will contribute to the development of effective patient-centered healthcare strategies.

In this study, 17 predictor variables were initially considered. However, following preliminary analyses, only eight statistically significant variables were included in the regression model to minimize the risk of overfitting. While the selected variables were appropriate for the sample size in this study, including additional significant variables without a corresponding increase in sample size could potentially elevate the risk of overfitting. To address this limitation, future studies should aim to secure a larger sample size or utilize advanced statistical techniques, such as cross-validation, to improve model stability and enhance generalizability.

This study sampled patients undergoing hemodialysis within a specific region, limiting the results’ generalizability. Additionally, the study’s cross-sectional nature limits the ability to establish clear causal relationships between variables, and reliance on self-reported surveys may result in discrepancies between reported and actual self-care behaviors. Therefore, future research should enhance the sample’s representativeness through multi-institutional studies across various regions and healthcare facilities and use longitudinal study designs to verify causal relationships. It would also be beneficial to enhance the reliability and validity of the data using objective indicators and medical records. As previously noted, online video-based education has been shown to overcome temporal and spatial limitations while enhancing knowledge acquisition. Additionally, the teach-back method has proven effective in improving comprehension. By developing an educational tool that integrates online video-based education with the teach-back method, it is possible to promote patient engagement in treatment and improve health outcomes. Future research should evaluate the long-term effects of these educational interventions and explore their applicability in diverse populations.

## 5. Conclusions

This study examined the effects of shared decision-making, health literacy, and self-care knowledge on the self-care behaviors of hemodialysis patients. A survey was conducted among 108 patients in the dialysis units of three medical institutions in Cheongju City, Korea. The findings demonstrated that health literacy is the most significant factor influencing self-care behaviors, with self-care knowledge also playing a crucial role in enhancing these behaviors. These results underscore the importance of customized educational programs to improve health literacy and self-care knowledge. Expanding the role of nurses in providing explanatory care is essential, along with adjusting the nurse-to-patient ratio in dialysis units to enable more individualized attention and care. National policies should focus on increasing nursing staff and establishing robust financial support systems to ensure patients have access to necessary resources. Moreover, systematic and continuous educational programs that provide patients with information and motivation are vital for further enhancing self-care capabilities. Comprehensive interventions incorporating shared decision-making, health education, and financial support are expected to maximize the self-care capabilities of hemodialysis patients, ultimately leading to improved treatment outcomes. 

## Figures and Tables

**Figure 1 healthcare-13-00175-f001:**
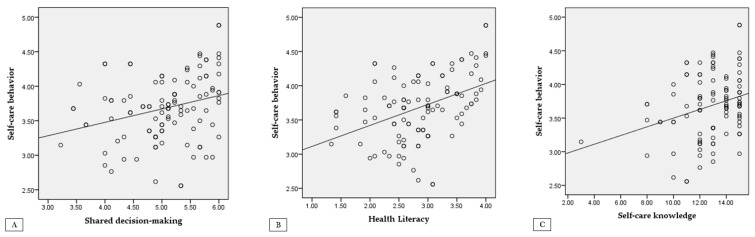
Correlations among the variables. (**A**) Correlation between self-care behavior and shared decision-making. (**B**) Correlation between self-care behavior and health literacy. (**C**) Correlation between self-care behavior and self-care knowledge.

**Table 1 healthcare-13-00175-t001:** Descriptive statistics of participant characteristics (N = 108).

Variables	Categories	*n* (%)	M ± SD	Min~Max
Sex	Men	56 (51.9)		
	Women	52 (48.1)		
Age (years)	<55	32 (29.6)	61.99 ± 13.81	26.00~88.00
	55~69	41 (38.0)		
	≥70	35 (32.4)		
Education	≤9th grade	36 (33.3)		
	10~12th grade	41 (38.0)		
	≥College level or higher	31 (28.7)		
Religion	Yes	52 (48.1)		
	No	56 (51.9)		
Caregiver	Yes	102 (94.4)		
	No	6 (5.6)		
Employment	Yes	32 (29.6)		
	No	76 (70.4)		
Economic status	Adequate	15 (13.9)		
	Moderate	64 (59.2)		
	Inadequate	29 (26.9)		
Dialysis education	Yes	108 (100.0)		
	No	0 (0.0)		
Dialysis Frequency (week)	2/week	2 (1.9)		
	3/week	106 (98.1)		
Dialysis duration (year)	≤3.0	36 (33.3)	6.53 ± 5.30	1.00~30.00
	3.1~8.9	41 (38.0)		
	≥9.0	31 (28.7)		
Duration of each session (hours)	4	108 (100)		
Cause of ESRD	Diabetes	57 (52.8)		
	Hypertension	26 (24.1)		
	Glomerulonephritis	11 (10.2)		
	Polycystic kidney disease	9 (8.3)		
	Others	5 (4.6)		
Number of comorbidities	1	26 (24.1)	2.18 ± 0.89	1.00~4.00
	2	44 (40.7)		
	3	30 (27.8)		
	4	8 (7.4)		
Type of comorbidities *	Hypertension	92 (85.2)		
	Diabetes	69 (63.9)		
	Cardiovascular disease	40 (37.0)		
	Malignancy	14 (13.0)		
	Liver disease	10 (9.3)		
	Others	8 (7.4)		
	Pulmonary disease	4 (3.7)		
Number of medications	≤8.9	40 (37.0)	10.29 ± 3.53	4.00~22.00
	9.0~12.9	44 (40.8)		
	≥13.0	24 (22.2)		

Notes: M, mean; SD, standard deviation; Min, minimum; Max, maximum; ESRD, end-stage renal disease. * Multiple response.

**Table 2 healthcare-13-00175-t002:** Descriptive statistics of shared decision-making, health literacy, self-care knowledge, and self-care behavior (N = 108).

Variables	Items	M ± SD	Min~Max	Scale Standardized Score		
				M ± SD	Min~Max	Range
Self-care behavior	35	125.28 ± 16.25	87.00~171.00	3.68 ± 0.48	2.56~4.89	1~5
Medication management	2	9.39 ± 1.13	6.00~10.00	4.69 ± 0.56	3.00~5.00	1~5
Vascular care	6	25.83 ± 3.70	14.00~30.00	4.31 ± 0.62	2.33~5.00	1~5
Physical care	11	43.85 ± 5.57	30.00~55.00	3.99 ± 0.51	2.73~5.00	1~5
Exercise and rest	4	13.93 ± 3.27	7.00~20.00	3.48 ± 0.82	1.75~5.00	1~5
Blood pressure and weight management	3	9.74 ± 3.36	3.00~15.00	3.25 ± 1.12	1.00~5.00	1~5
Dietary management	6	18.60 ± 3.90	6.00~24.00	3.10 ± 0.66	1.20~4.80	1~5
Social activities	3	7.00 ± 2.85	3.00~15.00	2.33 ± 0.95	1.00~5.00	1~5
Shared decision-making	9	45.65 ± 6.19	29.00~54.00	5.07 ± 0.69	3.22~6.00	1~6
Health literacy	12	34.36 ± 7.85	16.00~48.00	2.86 ± 0.65	1.33~4.00	1~4
Self-care knowledge	15	12.86 ± 2.10	3.00~15.00	0.86 ± 0.14	0.20~1.00	0~1

Notes: M, mean; SD, standard deviation; Min, minimum; Max, maximum.

**Table 3 healthcare-13-00175-t003:** Differences in hemodialysis self-care behaviors based on participant characteristics (N = 108).

Variables	Categories	M ± SD	t/F	*p* (Scheffé)
Sex	Men	3.58 ± 0.57	−2.39	0.019
	Women	3.79 ± 0.33		
Age (years)	<55 ^a^	3.44 ± 0.47	6.53	0.002 (a < b, c)
	55~69 ^b^	3.74 ± 0.39		
	≥70 ^c^	3.83 ± 0.51		
Educational status	≤9th grade	3.66 ± 0.42	0.93	0.400
	10~12th grade	3.63 ± 0.45		
	≥College level or higher	3.78 ± 0.57		
Religion	Yes	3.72 ± 0.60	0.85	0.399
	No	3.65 ± 0.52		
Caregiver	Yes	3.69 ± 0.48	0.64	0.526
	No	3.56 ± 0.49		
Employment	Yes	3.66 ± 0.55	−0.32	0.748
	No	3.69 ± 0.45		
Economic status	Adequate ^a^	4.00 ± 0.52	11.92	<0.001 (a > b, c)
	Moderate ^b^	3.75 ± 0.39		
	Inadequate ^c^	3.38 ± 0.47		
Dialysis duration	≤3.0	3.67 ± 0.58	1.63	0.200
	3.1~8.9	3.78 ± 0.37		
	≥9.0	3.58 ± 0.46		
Cause of ESRD	Diabetes	3.70 ± 0.46	1.38	0.245
	Hypertension	3.65 ± 0.52		
	Glomerulonephritis	3.45 ± 0.57		
	Polycystic kidney disease	3.92 ± 0.35		
	Others	3.82 ± 0.15		
Number of comorbidities	1	3.87 ± 0.36	1.76	0.160
	2	3.63 ± 0.60		
	3	3.38 ± 0.47		
	4	3.62 ± 0.37		
Number of medications	≤8.9	3.72 ± 0.58	0.74	0.478
	9.0~12.9	3.62 ± 0.33		
	≥13.0	3.74 ± 0.53		

Notes: M, mean; SD, standard deviation; ESRD, end-stage renal disease. ^a, b, c^ comparison groups of Scheffé test.

**Table 4 healthcare-13-00175-t004:** Factors affecting self-care behavior (N = 108).

Variable	B	SE	*β*	t	*p*
(Constant)	1.37	0.35		3.88	<0.001
Sex (ref. = women)					
Men	−0.25	0.07	−0.26	−3.54	0.001
Age	0.01	0.00	0.36	4.66	<0.001
Economic status (ref. = inadequate)					
Moderate	0.17	0.08	0.17	1.97	0.051
Adequate	0.35	0.12	0.25	2.82	0.006
Shared decision-making	0.03	0.06	0.05	0.58	0.565
Health literacy	0.30	0.07	0.41	4.55	<0.001
Self-care knowledge	0.04	0.02	0.17	2.15	<0.001
F (*p*)	13.59 (<0.001)				
Adjusted R^2^ (%)	45.2				
Tolerance	0.64~0.94				
VIF	1.06~1.57				
Durbin–Watson	2.51				

Notes: B, unstandardized coefficients; SE, standard error; *β*, standardized coefficients; adjusted R^2^, adjusted R square; VIF, variance inflation factors; ref., reference.

## Data Availability

The original contributions presented in the study are included in the article; further inquiries can be directed to the corresponding authors.

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
