# Peer review of "Effects of Shared Decision-Making, Health Literacy, and Self-Care Knowledge on Self-Care Behavior Among Hemodialysis Patients in Korea: A Cross-Sectional Survey"

_healthcare, 2025, doi:10.3390/healthcare13020175_

Round 1
Reviewer 1 Report
Comments and Suggestions for Authors
I was glad to review the authors' work regarding this interesting manuscript on effects of Shared Decision-Making, Health Literacy, and Self-Care Knowledge on Self-Care Behavior among Hemodialysis Patients in Korea.
The article is well-structured and addresses a relevant topic in the management of hemodialysis patients. However, minor revisions are necessary to improve the clarity of the methodology, data analysis, and graphical presentation. Additionally, greater attention to the study’s limitations and concrete practical suggestions would enhance the practical impact of the results. I can consider the paper for publication after minor changes.
Introduction
The introduction provides a clear overview of the issue, but some references are redundant. Reduce the repetition of the same information in subsequent paragraphs: Some concepts in the introductory section are repeated later in the text (e.g., effects of self-care knowledge). It is recommended to consolidate this information to avoid redundancy.
Methodology
The sample is representative of only a specific geographical area (C city, Korea): I recommend explaining why only these three hospitals were chosen to clarify the sampling method better.
Study Design
As the study is cross-sectional, it is not possible to establish causal relationships between the variables. It is recommended to consider a longitudinal follow-up to monitor the evolution of the variables over time.
Statistical Analysis
It is positive that the sample size calculation was performed. However, further justification for the choice of 17 predictive variables would be helpful: The rationale for including the predictive variables in the model should be further elaborated to avoid the risk of overfitting.
Tables and Figures
The data presentation is clear, but some tables (e.g., Tables 3 and 5) are particularly dense. Highlighting significant values in bold or using alternative formats would be helpful.
No graphs were included, which could make the interpretation of significant differences and correlations more immediate: consider to add a graph (of your choice) would help better visualize the differences in self-care behaviors and the observed correlations.
Discussion and limits
The discussions refer to the central role of nurses in the shared decision-making process. Nurses can assist patients in decision making by providing the necessary information and support: the relational skills of nurses in this particular setting are indispensable as described in a very recent review study (https://doi.org/10.1016/j.nepr.2024.104229 ); a clarification in this sense could be useful.
The discussion is comprehensive but could be improved by better highlighting the study’s limitations and suggesting practical approaches based on the results: Expanding the limitations section and proposing concrete recommendations for educational interventions based on the findings is advised.
Conclusions
The conclusion summarizes the main results well but could include more specific practical recommendations: In the conclusions, it would be helpful to provide more concrete practical recommendations to improve financial support and health education for hemodialysis patients.
References
The formatting of the references does not appear to comply with the journal's guidelines. For instance, article citations require the following format:
“Author 1, A.B.; Author 2, C.D. Title of the article. Abbreviated Journal Name Year, Volume, page range.”
Please review and adjust where necessary.
Author Response
We appreciate the time and effort you and the reviewers have put into providing valuable feedback and insightful comments, improving our manuscript. We have carefully considered each comment and updated the manuscript as required. We have marked the revisions made to the manuscript in red font. The italicized text in this response indicates the revised content incorporated into the manuscript.

Reviewer 2 Report
Comments and Suggestions for Authors
This manuscript is very informative and need some minor revision.
Abstract: no need to change
Introduction section: well explained and no need to change
Methods section: it is better that N:5 in line 85 delete and add on exclusion criteria like this : those with cognitive, physical and mental complication requiring specialize treatment.
Line 100: you can change word helper to assistant.
Line 103- 104: it is better to changes chronic diseases to chronic disorders.
Line 136: it is better to changes artificial kidney room to hemodialysis center or unite.
Line 159: I think that high school level is different between countries.
Education in table 1 need to change.
I think that you can classify No income patients in low economic group.
Between lines 160- 162 pointed that 70.4 % of participant had no income, but the majority of participant felt that their economic status was moderate. I think it is better that participant with no income to be comforted in inadequate economic status.
Author Response

(The authors gave the same response as above.)
